# Factors Determining the Occurrence of Frailty Syndrome in Hospitalized Older Patients

**DOI:** 10.3390/ijerph191912769

**Published:** 2022-10-06

**Authors:** Izabela Kozicka, Agnieszka Guligowska, Joanna Chrobak-Bień, Katarzyna Czyżewska, Natalia Doroba, Anna Ignaczak, Anna Machała, Ewelina Spałka, Tomasz Kostka, Ewa Borowiak

**Affiliations:** 1Department of Conservative Nursing, Faculty of Health Sciences, Medical University of Lodz, Jaracza 63, 90-251 Lodz, Poland; 2Department of Geriatrics, Faculty of Health Sciences, Medical University of Lodz, Plac Hallera 1, 90-647 Lodz, Poland

**Keywords:** frailty syndrome (FS), functional ability (FA), handgrip strength (HGS), depression, elderly people

## Abstract

Frailty syndrome (FS) is a condition characterized by a decline in reserves, observed with aging. The most important consequences of the frailty syndrome include disability, hospitalization, fractures, institutionalization, and early mortality. The aim of this study was to identify the most important risk factors for FS in a group of older hospitalized patients in Poland. A total of one hundred and forty-one (78 women, 63 men) elderly patients from the Departments of Internal Medicine of the Medical University of Lodz (Poland) were recruited for this study. Frailty Instrument of the Survey of Health, Aging and Retirement in Europe (SHARE-FI), handgrip strength (HGS), depressive symptoms using the Geriatric Depression Scale (GDS), and functional ability (FA) using the Instrumental Activities of Daily Living (IADL) were assessed. According to SHARE-FI score, participants were divided into control group, frail, and pre-frail patients. Out of all 141 tested patients, FS was confirmed in 55 patients, and pre-frailty was observed in 52 patients. The occurrence of FS in the group of studied patients was related to age (*p* < 0.001), widowhood (*p* < 0.001), comorbidities (*p* < 0.001), heart diseases (*p* = 0.04), more medications taken (*p* < 0.001), lower FA (*p* < 0.001), weaker HGS, and depression (*p* < 0.001). The strongest positive correlations were between Share-FI score and the number of diseases (rS = 0.31), GDS (rS = 0.32), while negative correlations with IADL (rS = −0.47) and HGS (rS = −0.35). The study shows that FS is associated with age, comorbidities, number of medications taken, and widowhood. The present study has also demonstrated that FA, depression, and especially HGS are essential determinants of FS of elderly hospitalized people.

## 1. Introduction 

In recent years, the identification of the elderly with frailty syndrome (FS), or those at risk of developing FS has become a cornerstone of geriatric care [1]. Due to the lack of a uniform definition, there are serious difficulties in diagnosing this syndrome, which significantly hinders the daily functioning of seniors. According to the definition proposed by L. P. Fried, it is a physiological syndrome characterized by a decrease in reserves and resistance to stressors [2]. In this classical definition, FS includes parameters such as reduced muscle strength (handgrip strength less than 20% for gender norm and body mass index (BMI)), subjective fatigue, unintentional weight loss (at least 4.5 kg per year), slowed walking (less than 20% of the time-to-walk gender 4.572 m norm), and low physical activity. Three of the above-mentioned criteria are sufficient for making a diagnosis. FS is usually preceded by prefrailty syndrome, which includes one or two symptoms, and identifies a group of elderly people with a significantly increased risk of developing FS [3]. Patients diagnosed with FS deserve special attention because they have an increased risk of disability, which leads to frequent hospitalizations, worse prognosis after surgery, decreased response to treatment, and an increased risk of death [4,5,6,7]. FS contributes to excessive costs of health care, polypharmacy, hospitalization, and institutionalization of the elderly [8,9,10,11,12,13,14]. FS causes a loss of independence and thus also presents enormous challenges for families, carers, and other welfare structures [15].

FS may be conceptually defined as a clinically recognizable state in older people who have increased vulnerability, resulting from age-associated declines in physiological reserve and function across multiple organ systems, such that the ability to cope with every day or acute stressors is compromised [1,2,5,16,17]. 

The prevalence of FS in people over 65 in Europe ranges from 5.8% to 27.3%; in addition, from 34.6% to 50.9% are classified as "pre-frail" [15]. Lodz is the fastest aging city in Poland. In 2023, people over 65 will make up a quarter of Lodz’s population, and by 2050 they will account for 37%. Therefore, we will deal with the so-called "double aging", that is, the rapid increase in the number of very old people. Such a demographic situation generates the need for appropriate medical and environmental care for elderly people [18]. It is worth mentioning that in this age group the frequency of FS is around 10%, and even up to 50%. The percentage of people with FS grows proportionally with age. For people aged 80–89, it is about 20%, and 33.3% in patients 90+ [19]. With the rapid aging of the population, FS will reach epidemic proportions in the next few decades. That is why the main aim of geriatric care in Poland and worldwide is, above all, to maintain a high level of functional ability (FA) and independence in life, to provide treatment and rehabilitation, to improve quality of life, and to prevent FS [20,21,22].

Therefore, the objective of this study was to identify the most important risk factors for FS in a group of elderly patients hospitalized in Polish Departments of Internal Medicine. In particular, the relationship between FS and HGS, depression, FA was analyzed.

## 2. Materials and Methods

### 2.1. Patients

This study was conducted as part of obligatory practical classes in geriatric nursing with students in a strictly defined period of the academic year from March to June 2022. Out of all elderly patients staying, at that time, in three Departments of Internal Medicine of the Medical University of Lodz, Poland, 141 (63 men and 78 women) elderly patients were selected who met the inclusion criteria of the study. The study covered patients suffering from various chronic conditions: high blood pressure or hypertension; diabetes or high blood sugar; chronic lung disease, such as chronic bronchitis or emphysema; asthma; osteoporosis; rheumatic disease; stroke victims; and others. The inclusion criteria of the study were as follows: (*i)* aged over 65 years, (*ii)* good verbal and logical communication (without severe dementia), (*iii)* no considerable hearing disorders significantly hampering understanding of questions at the time of the study. All patients staying in the hospital during the specified period and meeting the above criteria were included in the study. Each subject underwent a multidimensional assessment, which included: demographic and social parameters, health conditions, and physical function. The following set of instruments commonly used for health assessments of elderly populations was utilized: functional abilities (FA), depressive symptoms (15-item Geriatric Depression Scale, GDS), handgrip strength (HGS), and Frailty Syndrome (FS) (SHARE-FI questionnaire). 

### 2.2. Measurements

#### 2.2.1. Frailty Syndrome

FS was evaluated using the Frailty Instrument of the Survey of Health, Aging and Retirement in Europe (SHARE-FI). SHARE-FI is especially recommended in primary health care and hospital care in elderly people [2,23]. Translation and validation procedure of the Polish version of the SHARE-FI was completed by Muszalik [24]. The questions included in this procedure cover the following areas: sex of the subject, feeling of exhaustion, loss of appetite, difficulty in walking upstairs, reduction in physical activity, and assessment of HGS [24,25]. The obtained results, calculated with the use of the SHAREFI calculator, qualify the examined person into one of the three groups: non-frail, pre-frail, and frail [23,25]. Qualification of the subject as frail: score > 3 for men and >2.13 for women; pre-frail: 1.21–3 for men and 0.32–2.13 for women; and non-frail: <1.21 for men and <0.32 for women [15].

#### 2.2.2. Handgrip Strength Measurements

Handgrip strength (HGS) was tested using a hydraulic hand dynamometer by Jamar^®^ (Sammons Preston Rolyan, Mississauga, ON, Canada). The dynamometer facilitates measurements of isometric force to 90 kg. According to the literature, measurements are influenced by subject position when carrying out the test (lying, sitting, or standing). Scientific evidence has shown better results with a standing position. Each of the participants carried out the handgrip strength test while standing with his/her shoulder adducted and neutrally rotated, and with the elbow in 90 degrees flexion with no radioulnar deviation. The measurements were performed two times each for the right and left hand with pauses between measurements. The results were recorded as kilogram force [26]. The better result of the dominant hand was used for the analysis.

#### 2.2.3. Functional Ability Test

Using the Instrumental Activities of Daily Living scale (IADL), the ability to perform complex activities of daily living, such as using the phone, shopping, preparing meals, cleaning, laundry, use of different means of transport, and self-managing medication and handling money were evaluated. The total score is relevant to a particular patient and a fall in the score on consecutive examinations reflects a deteriorated general state. The result of the activity of daily living assessment of an elderly person on this scale allows for the objectivization of the patient’s needs for care or necessary assistance. The maximum number of points that can be obtained on this scale is 27 [27].

#### 2.2.4. Emotional Status Assessment

Evaluation of the participants’ emotional status was undertaken using the GDS, characterizing the state of well-being of the subject in the previous two weeks. The GDS contains 15 questions characterizing a person’s depressive status. Scores ranging from 0 to 5 indicate normal mood; scores between 5 and 9 indicate a risk of depressive symptoms; and scores > 9 indicate severe depressive symptoms [28].

#### 2.2.5. Statistical Analysis

The normality of distribution was verified using the Shapiro–Wilk test. All the continuous variables were not normally distributed, therefore they were presented by median and interquartile difference (from the first (25%) to the third (75%) quartile). The quantitative variables (between the sexes), were compared using the Mann–Whitney U-test. The occurrence of differences between the 3 groups was assessed using a one-way ANOVA, Kruskal–Wallis test and the chi-square test or Fischer’s exact test. Spearman correlations between SHARE-FI score and numerical data were calculated. Logistic stepwise regression (odds ratios and corresponding 95% confidence intervals (95% CI)) was used to assess which independent variables predicted the presence of FS. The following variables were taken into account when building the model: age, marital status, heart diseases, number of diseases, IADL, GDS, and HGS max. The model was adjusted for sex and BMI. Statistical significance was set at *p* ≤ 0.05. The analyses were performed using Statistica 13.1 (StatSoft Polska, Cracow, Poland).

#### 2.2.6. Ethical Considerations

The proposal of this study did not require the approval of the Bioethics Committee, as it does not bear the hallmarks of a medical experiment. The study was conducted in accordance with the guidelines of the Helsinki Declaration. Patients signed informed consent for all the diagnostic and therapeutic procedures during hospitalization. All of the gathered data were confidential.

## 3. Results

Table 1 presents the general characteristics of the study population according to sex. The median age was 73 for both sexes. The Mann–Whitney U test showed that the average HGS for women was 20 kg, while the mean HGS for men was 30 kg. The median frailty score was 2.02 for all participants, whereas the average frailty score in women was 2.22, while the mean frailty score for men was 1.98. Age of the subjects, number of diseases, number of medications taken, BMI, IADL, GDS, and frailty score did not differ significantly. The prevalence of diseases was also similar in both sexes, with the exception of heart diseases and arthritis which were more common in women (*p* < 0.05) and respiratory system diseases which were more common in men (*p* < 0.01).

Comparison of anthropometric variables, test results, and prevalence of chronic diseases between frail, pre-frail, and control group are presented in Table 2. Among the patients diagnosed with FS based on the SHARE-FI scale, the median age was 77 years, higher than in the pre-frail group–71.5 years, and control group–69.5 years, and was statistically significant (*p* < 0.01). Patients with FS had a statistically significantly higher number of diseases and a higher number of medications taken compared to patients with pre-frail and the control group (*p* < 0.001). The most common diseases in the subjects with FS were heart diseases (*p* < 0.05). The other diseases were not statistically significant. There were no significant differences between body mass, height, BMI of the subjects and the frequency of FS occurrence. A lower level of FA, weaker HGS, and a greater number of depressive symptoms were demonstrated by patients with frailty compared to the pre-frail as well as the control group (*p* < 0.001). People with frailty were also significantly more often widowed, while pre-frail patients and with the control group were married (*p* < 0.01). To improve the visualization of these obtained relationships, examples are presented in the form of box plots (Figure 1).

Spearman’s rank correlation coefficient between selected variables and SHARE-FI score are presented in Table 3. Frailty score was related to almost all quantitative variables that are presented, excluding body mass, height, and BMI of all participants. Frailty score correlated negatively with IADL and HGS, whereas it correlated positively with age of the subject, number of diseases, number of medications taken, as well as GDS. In the group of women, frailty score correlated negatively with IADL and HGS, and positively with age, number of diseases, number of medications taken, and GDS. In the group of men frailty score correlated negatively with IADL and HGS, and positively with GDS. Generally, the strongest correlations were noted between frailty score and number of diseases and IADL in women while the strongest correlations were noted between frailty score and IADL and HGS in men. To improve the visualization of these obtained relationships, they are presented in the form of a graph (Figure 1). The relationship between handgrip strength max. (HGS max.) and frailty score in men and women is given in Figure 2.

Odds ratios obtained in one-factor analysis are presented in Table 4. 

Among other things, the obtained data show that the odds of FS increase by 6% with each year of life. In addition, in the presence of heart disease in the elderly, the odds of developing FS rises by 150%. However, each subsequent coexisting disease increases the risk of FS by 40%. The chances of FS occurrence are reduced by HGS (each additional kilogram by 10%) and IADL (each additional point by 18%).

Additionally, all these statistically significant factors and obtained differences in previous analyses were confirmed by logistic regression. The following were statistically significant in the model: HGS (OR = 0.91 (0.86–0.96), *p* < 0.001); number of diseases (OR = 1.5 (1.2–1.96), *p* < 0.001) IADL (OR = 0.87 (0.79–0.96), *p* = 0.004). Heart diseases, age, sex, BMI, GDS, and marital status (widowed) did not enter the model. The ROC (Receiver Operating Characteristics) curve was generated for the model, the area under the AUC (Area Under The Curve) curve is 0.84, and the AUC error = 0.032, which proves the high correctness of the classification.

## 4. Discussion

FS is a serious problem that concerns elderly people. It contributes to the deterioration of the quality of life of seniors at all levels of their functioning [5]. It is a multifaceted problem that results from the disturbed regulation of many organs and systems involved with the aging of the body. It presents itself by numerous clinical problems, has a varied course, and is associated with an increased risk of complications [29]. In the event of the appearance of FS, the risk of hospitalization of an elderly person increases, as well as the risk of losing independence, susceptibility to more frequent illness, and even death [30].

The results of this study, conducted with 141 elderly hospitalized people, showed that the prevalence of pre-frailty syndrome was 36.9% and the level of actual suffering from FS was 39%. The available data in the literature are, in most cases, consistent with the results obtained in this research, and prove that FS is a common problem for the aging population in Poland, as well as worldwide [31]. When it comes to the prevalence of FS, the highest results were recorded in Europe, including in Poland, but also in Italy and Spain [32]. 

Our study did not show any relationship between the occurrence of the FS and the gender of the respondents, similar to the study conducted in Poland by Muszalik et al. [33]. The occurrence of the FS in the conducted study was determined by age of the subject and marital status. Widowed people were characterized by frailty or pre-frailty, while those in the control group were married. Several studies have also shown that the prevalence of FS increases with age [31,33,34]. However, widowhood and the number of comorbidities were also risk factors for the occurrence of FS in the studies conducted by Liu et al. [34,35]. The present study confirmed that people in the pre-frail and frail groups took more medications per day compared to those in the control group. Interesting findings were also presented by Saum et al. in Germany. Based on an 11-year prospective study, it was confirmed that the use of large amounts of drugs increases the risk of developing the FS, regardless of the number of comorbidities [36].

The present study found a relationship between the prevalence of the FS and FA, as well as emotional status among hospitalized patients. Seniors with FS were much more often characterized by declining FA and suffering from depression. In a cross-sectional study conducted by Thinuan et al. the existence of such a relationship was confirmed, especially in developed countries [34]. Research conducted by Firuzan et al. confirms that depression may have an important role in the development of FS in both sexes [37]. Furthermore, in the study of Boyer et al. a large part of cohort was frail or pre-frail and presented signs of loss of independence, which may be explained by multiple factors including poor FA and depression [38]. Thinaun et al., in their research, also showed that one of the risk factors affecting incidence of FS in developed and developing countries are cardiovascular diseases [34]. A similar relationship was demonstrated in these studies. The most common diseases in the subjects with the FS were heart diseases. In the women participating in our study, the strongest correlations were noted between the frailty score and age, number of diseases, and IADL. These results confirm that FS in women may be associated with older age, multi-disease, and worse FA. On the other hand, in the men, the strongest correlations were observed between the frailty score and IADL and HGS, which means that the occurrence of the FS in men is strongly associated with both worse FA and weak HGS. HGS is part of the FI - therefore the correlation between them is obvious, but it should be underlined that in multivariate analysis HGS stays significant. Interestingly, in men, there was no association between FS and age or the number of comorbidities. The strongest association was observed with FS, especially with HGS in the men, whereas in the women there was no such a strong relationship. Depression also turned out to be, to a certain extent, a determinant of FS in both sexes, which is also confirmed by many available scientific reports [34,37,38,39,40,41]. Our research has shown that widowhood is one of the risk factors for FS. To the best of our knowledge, studies assessing the relationship between FS and widowhood are not very often conducted. In the available literature, there are studies that show that even the least frail widowed individual has higher mortality than a married person of the same age and sex. An 85+ year old widowed man is expected to have double the risk of dying compared to a married man of the same age [42].

## 5. Conclusions

In conclusion, the present study has demonstrated that age, comorbidities, in particular cardiovascular diseases, polypharmacy, and widowhood are risk factors of FS in elderly people hospitalized in the Departments of Internal Medicine. Additionally, we showed that depression, decrease in FA, and weak HGS are the most important determinants for FS development in elderly patients. Our results confirm that FS in women is associated with older age, multi-disease, and worse FA. However, the occurrence of FS in men is strongly related to both worse FA and weak HGS. According to the obtained results, it seems that in hospitalized older adults, the occurrence of FS depends on multiple factors and may be important for further prognosis. An especially important aspect of FS prevention seems to be caring for muscle strength. Previous and present data show that low HGS is a crucial element of FS and should be included as a standard screening test for older adults’ care. Patients with a low score should be given special care and have a further diagnosis of frailty syndrome. Undoubtedly, further studies are needed to investigate the potential possibilities of alleviating the symptoms of FS.

## 6. Study Limitations 

This study has several limitations, including (1) the relatively small sample of participants, (2) inclusion of only the Caucasian race in the study, (3) no use of the gait speed method. It is worth emphasizing that the strength of this study is that it demonstrates a significant relationship between FS and poorer HGS. It points to the need to introduce HGS measurement in older adults as a recommended component of the Comprehensive Geriatric Assessment in medical facilities. Bearing in mind these limitations, the present study requires caution in data interpretation and should be continued and confirmed in future studies in larger groups of seniors.

## Figures and Tables

**Figure 1 ijerph-19-12769-f001:**
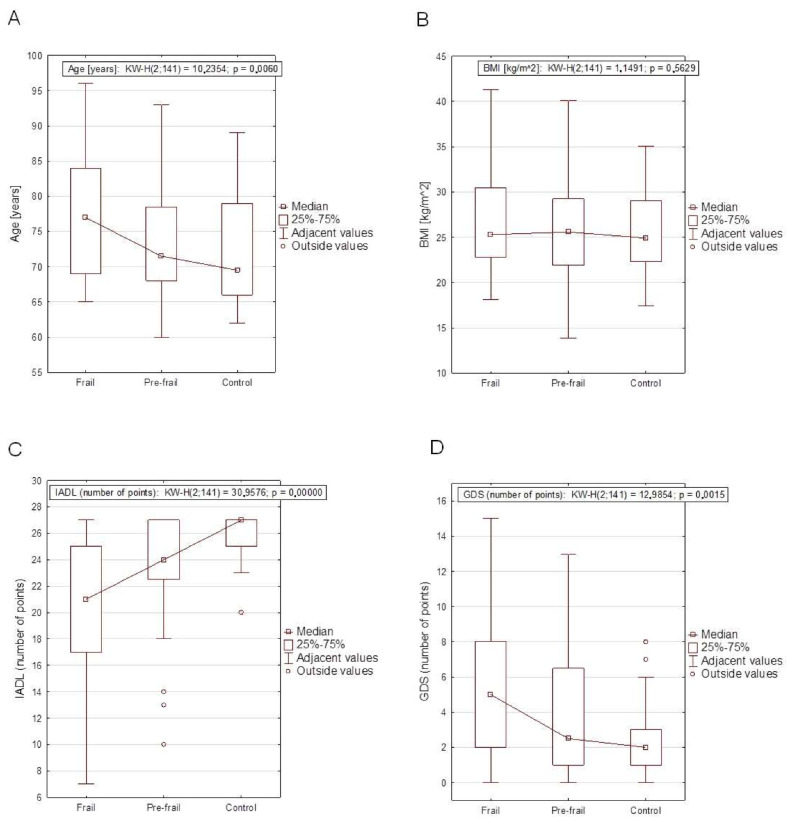
Box-plot graphs among Frail, Pre-frail and Control groups with the results of the Kruskal–Wallis test of age—(**A**); body mass index (BMI)—(**B**); instrumental activities of daily living (IADL)—(**C**); geriatric depression scale (GDS)—(**D**).

**Figure 2 ijerph-19-12769-f002:**
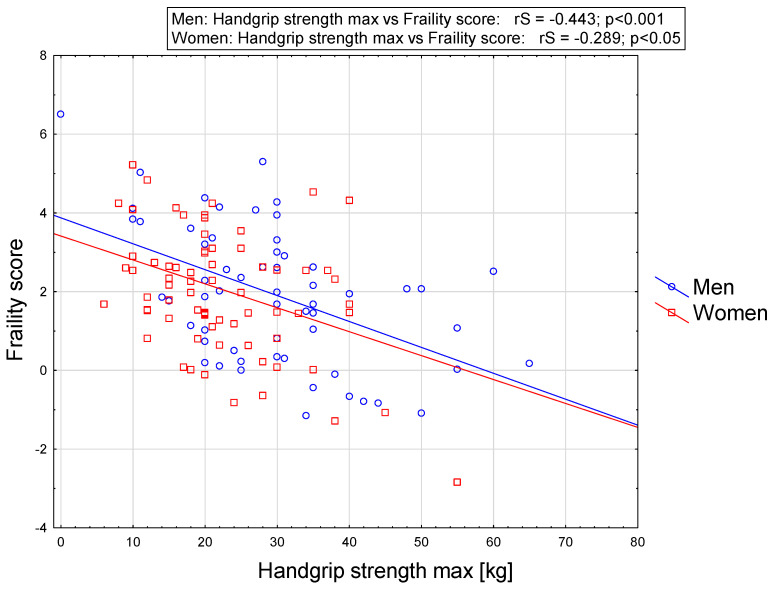
Relationship between HGS max. and frailty score in men and women.

**Table 1 ijerph-19-12769-t001:** General characteristics of the study population (*n* = 141) according to sex.

Quantitative Variables	All *n* = 141	Women *n* = 78	Men *n* = 63	Mann-Whitney U-Test/Chi2 *p*-Value
Demography				
Age [years]	73 (68–81)	73 (67–81)	73 (68–80)	ns
Education [*n*, %]				ns
Higher	21 (14.9)	10 (12.8)	11 (17.5)	
Secondary	61 (43.3)	35 (44.9)	26 (41.3)	
Vocational	31 (22)	19 (24.4)	12 (19.1)	
Primary	28 (19.9)	14 (18)	14 (22.2)	
Place of living [*n*, %]				ns
Urban	118 (83.7)	65 (83.3)	53 (84.1)	
Rural	23 (16.3)	13 (16.7)	10 (15.9)	
Living status [*n*, %]				ns
Alone	65 (46.1)	38 (48.7)	27 (42.9)	
With family	76 (53.9)	40 (51.3)	36 (57.1)	
Marital status [*n*, %]				ns
married	62 (44)	30 (38.5)	32 (50.8)	
not married/single	14 (9.9)	7 (9)	7 (11.1)	
widowed	65 (46.1)	41 (52.6)	24 (38.1)	
Economic status [*n*, %]				ns
very good	9 (6.4)	4 (5.1)	5 (7.9)	
good	56 (39.7)	36 (46.2)	20 (31.8)	
average	75 (53.2)	38 (48.7)	37 (58.7)	
bad	1 (0.7)	0	1 (1.6)	
Anthropometry and state of health				
BMI [kg/m^2^]	25.3 (22.6–29.7)	25.3 (22.8–29.3)	24.9 (21.8–32)	ns
IADL [number of points]	24 (20–27)	24 (20–26)	24 (21–27)	ns
IADL Category [*n*, %]				ns
low	2 (1.4)	1 (1.3)	1 (1.6)	
medium	26 (18.4)	14 (18)	12 (19.1)	
high	113 (80.1)	63 (80.8)	50 (79.4)	
Mobility [*n*, %]				ns
independent	108 (76.6)	59 (76)	49 (77.8)	
cart	3 (2.1)	2 (2.6)	1 (1.6)	
balcony	7 (5)	5 (6.4)	2 (3.2)	
crutches	16 (11.3)	8 (10.3)	8 (12.7)	
lying patient	7 (5)	4 (5.1)	3 (4.8)	
HGS max. [kg]	22 (18–30)	20 (15–28)	30 (20–35)	0.0004
GDS [number of points]	3 (1–7)	3 (1–6)	2 (1–7)	ns
GDS Category [*n*, %]				ns
normal mood	104 (73.8)	59 (75.6)	45 (71.4)	
risk of depressive symptoms	26 (18.4)	14 (18)	12 (19.1)	
severe depressive symptoms	11 (7.8)	5 (6.4)	6 (9.5)	
Fraility score	2.02 (1.02–3.1)	2.22 (1.27–3.02)	1.98 (0.5–3.2)	ns
Diseases				
Number of diseases	3 (2–5)	3 (2–5)	3 (2–5)	ns
Number of medications taken	6 (3–8.5)	6 (3–9)	6 (3–8)	ns
Use of a hearing aid	19 (13.5)	7 (9)	12 (19.1)	ns
Use of glasses	112 (79.4)	61 (78.2)	51 (81)	ns
Heart diseases, *n* [%]	42 (29.8)	29 (38.2)	13 (20.6)	0.03
Hypertension, *n* [%]	85 (60.3)	52 (66.7)	33 (52.4)	ns
High blood cholesterol, *n* [%]	20 (14.2)	8 (10.4)	12 (19.1)	ns
Stroke, *n* [%]	9 (6.4)	6 (7.7)	3 (4.8)	ns
Diabetes, *n* [%]	42 (29.8)	22 (28.2)	20 (31.8)	ns
Respiratory system diseases, *n* [%]	17 (12.1)	4 (5.1)	13 (20.6)	0.005
Cancer, *n* (%)	12 (8.5)	5 (6.4)	7 (11.1)	ns
Stomach or duodenal ulcer, *n* [%]	9 (6.4)	6 (7.7)	3 (4.8)	ns

***Notes****:* Mann–Whitney U-test/Chi2/Fischer’s exact test were calculated. *Abbreviations:* BMI, body mass index; IADL, instrumental activities of daily living; HGS, handgrip strength (kg); GDS, geriatric depression scale; ns, statistically non-significant difference.

**Table 2 ijerph-19-12769-t002:** Comparison of anthropometric variables, test results, and prevalence of chronic diseases between Frail, Pre-frail, and Control group.

Quantitative Variables	Frail *n* = 55	Pre-frail *n* = 52	Control *n* = 34	Kruskal–Wallis *p*-Value
Demography				
Age [years]	77(69–84)	71.5 (68–78.5)	69.5 (66–79)	0.006 ^b^
Men [*n*, %]	19 (35)	24 (46.2)	20 (58.8)	ns
Education [*n*, %]				ns
Higher	11 (20)	6 (53.8)	4 (11.8)	
Secondary	17 (30.9)	28 (3.9)	16 (47.1)	
Vocational	12 (21.8)	11 (21.2)	8 (23.5)	
Primary	15 (27.3)	7 (13.5)	6 (17.7)	
Place of living [*n*, %]				ns
Urban	44 (80)	45 (86.6)	29 (85.3)	
Rural	11 (20)	7 (13.5)	5 (14.7)	
Living status [*n*, %]				ns
Alone	31 (56.4)	20 (38.5)	14 (41.2)	
With family	24 (43.6)	32 (61.5)	20 (58.8)	
Marital status [*n*, %]				0.006
married	14 (25.5)	29 (55.8)	19 (55.9)	
not married/single	7 (12.7)	6 (11.5)	1 (2.9)	
widowed	34 (61.8)	17 (32.7)	14 (41.2)	
Economic status [*n*, %]				ns
very good	5 (9.1)	4 (7.7)	0	
good	26 (47.3)	21 (40.4)	9 (26.5)	
average	24 (43.6)	26 (50)	25 (73.5)	
bad	0	1 (1.9)	0	
Anthropometry and state of health				
BMI [kg/m^2^]	25.3 (22.8–30.5)	25.6 (21.9–29.3)	24.9 (22.3–29.1)	ns
IADL [number of points]	21 (17–25)	24 (22.5–27)	27 (25–27)	<0.001 ^a.b.c^
IADL Category [*n*, %]				0.0075
low	2 (3.6)	0	0	
medium	17 (30.1)	7 (13.5)	2 (5.9)	
high	36 (65.5)	45 (86.5)	32 (94.1)	
Mobility [*n*, %]				0.019
independent	35 (63.6)	40 (76.9)	33 (97)	
cart	1 (1.8)	1 (1.9)	1(2.9)	
balcony	3 (5.5)	4 (7.7)	0	
crutches	10 (18.2)	6 (11.5)	0	
lying patient	6 (10.9)	1 (1.9)	0	
HGS max. [kg]	20 (13–25)	25 (20–32.5)	32 (24–42)	<0.001 ^a.b.c^
GDS [number of points]	5 (2–8)	2.5 (1–6.5)	2 (1–3)	0.0015 ^b^
GDS Category [*n*, %]				ns
normal mood	35 (63.6)	40 (77)	20 (85.3)	
risk of depressive symptoms	14 (25.5)	10 (19.2)	2 (5.9)	
severe depressive symptoms	6 (10.9)	2 (3.9)	3 (8.8)	
Fraility score	3.54 (2.69–4.15)	1.68 (1.42–2.11)	0.05 (−0.66–0.5)	0.0001 ^a.b.c^
Diseases				
Number of diseases	4 (3–6)	3 (2–3.5)	3 (2–5)	0.0001 ^b.c^
Number of medications taken	8 (6–10)	5 (2.5–7)	5 (3–7)	0.0003 ^b.c^
Use of a hearing aid	9 (16.4)	6 (11.5)	4 (11.8)	ns
Use of glasses	48 (87.3)	39 (75)	25 (73.5)	ns
Heart diseases, *n* [%]	23 (41.8)	13 (26)	6 (17.7)	0.039
Hypertension, *n* [%]	34 (61.8)	28 (53.9)	23 (67.7)	ns
High blood cholesterol, *n* [%]	11 (20)	4 (7.84)	5 (14.7)	ns
Stroke, *n* [%]	7 (12.7)	2 (3.85)	0	ns
Diabetes, *n* [%]	19 (34.6)	15 (28.9)	8 (23.5)	ns
Respiratory system diseases, *n* [%]	8 (14.5)	4 (7.7)	5 (14.7)	ns
Cancer, *n* [%]	6 (10.9)	5 (9.6)	1 (2.9)	ns
Stomach or duodenal ulcer, *n* [%]	5 (9.1)	1 (1.9)	3 (8.8)	ns

*Notes:* Kruskal−Wallis/Chi2/Fischer’s exact test were calculated. ^a^—Prefrail vs. control. ^b^—Frail vs. control. ^c^—Frail vs. Prefrail. *Abbreviations:* BMI, body mass index; IADL, instrumental activities of daily living; HGS, handgrip strength (kg); GDS, geriatric depression scale; ns, statistically non-significant differences.

**Table 3 ijerph-19-12769-t003:** Spearman’s rank correlation coefficient of quantitative variables and frailty score in men and women groups.

Quantitative Variables	Spearman’s Rank Correlation Coefficient in All	Spearman’s Rank Correlation Coefficient in Women	Spearman’s Rank Correlation Coefficient in Men
Age [years]	0.2384 **	0.3352 **	0.1217
Number of diseases	0.3089 ***	0.4565 ***	0.1417
Number of medications taken	0.2164 *	0.2723 *	0.1488
BMI [kg/m^2^]	0.1236	0.0335	0.2081
IADL [number of points]	−0.4686 ***	−0.3766 ***	−0.5642 ***
HGS max. [kg]	−0.3485 ***	−0.2891 *	−0.4430 ***
GDS [number of points]	0.3159 ***	0.2443 *	0.3810 **

*Notes:* Spearman’s rank correlation coefficient was calculated. Significant correlations (* *p* < 0.05, ** *p* < 0.01, *** *p* < 0.001). *Abbreviations:* BMI, body mass index; IADL, instrumental activities of daily living; HGS, handgrip strength (kg); GDS, geriatric depression scale.

**Table 4 ijerph-19-12769-t004:** The odds ratio of FS occurrence in one-way analysis for selected variables.

Quantitative Variables	Odds Ratio	95% Confidence Intervals	*p*-Value
Women (reference group)	1		
Men	0.50	0.25–1.01	ns
Age [years]	1.06	1.02–1.11	0.0036
Married (reference group)	1		
Not married/single	3.43	1.03–11.44	0.0451
Widowed	3.76	1.74–8.11	0.0007
Witout heart diseases (reference group)	1		
Heart diseases	2.46	1.17–5.16	0.0172
Number of diseases	1.41	1.17–1.69	0.0003
BMI [kg/m^2^]	1.04	0.98–1.11	ns
IADL [number of points]	0.82	0.75–0.90	0.0000
GDS [number of points]	1.13	1.03–1.25	0.0096
HGS max. [kg]	0.90	0.86–0.94	0.0000

***Notes:*** Odds ratios (ORs) and confidence intervals (CIs) with 95% confidence limits were calculated. *p*-values were highlighted in bold. *Abbreviations:* BMI, body mass index; IADL, instrumental activities of daily living; HGS, handgrip strength (kg); GDS, geriatric depression scale; ns, statistically non-significant differences.

## Data Availability

Not applicable.

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
