# Peer review of "Factors Determining the Occurrence of Frailty Syndrome in Hospitalized Older Patients"

_ijerph, 2022, doi:10.3390/ijerph191912769_

Round 1
Reviewer 1 Report
Thank you for the opportunity to review this manuscript where the objective of this study was to identify the most important risk factors for the FS in the group of elderly patients hospitalized in Polish Departments of Internal Medicine.
Section 2.1 - how are the patients approached and screened? Is it systematic sampling or convenience sampling? 141 seems a little strange as a total number with more females than males. what was the minimum sample size required to answer the main objective of the study?
Section 2.2.1 - what is the rationale behind using SHARE-FI versus other frailty measures such as CFS, FRAIL or Fried's phenotype? Surprised that gait speed was not measured for these patients as it is important indicator of frailty?
Section 2.2.6 - I am surprised that ethics approval was not required for this study and yet the authors mentioned that patients signed informed consent form before procedures were performed. This brings me to mention lines 331-332. There's a typo for line 331. So shouldn't there be a statement for line 332 since informed consent was taken?
Section 3 - I personally feel that there's not much added value to include table 1 as some of the comparisons made and results obtained are obvious especially height, weight and handgrip strength (eg. men would be taller, heavier and bigger handgrip strength than women). If table 1 is important, would suggest to remove height and weight and only include BMI. One surprising observation is that percentage of women with heart diseases are significantly higher than men. Is that a common trend in Poland or could it be due to sampling issue?
Table 2 makes more sense for me to answer the research question. Once again, would suggest to remove height and weight.
Table 3 - do this need to be included or can it be parked under Appendix?
Table 4 - would appreciate a conventional presentation of logistic regression analysis
Figure 1 - the resolution of the plots are not great and the caption within each plot is a little confusing, can it be clearer?
Section 4 - would want to see the strength and limitations of the study and clinical implications of the findings from this study.
Author Response
Response to Reviewer’s 1 Comments
We would like to thank the Reviewer for the analysis of this manuscript and the valuable remarks. The manuscript has been corrected according to the Reviewer’ comments. Below we provide a summary of our responses to the main remarks:
Point 1: Section 2.1 - how are the patients approached and screened? Is it systematic sampling or convenience sampling? 141 seems a little strange as a total number with more females than males. what was the minimum sample size required to answer the main objective of the study?
Response 1: Thank you very much for this comment. Total sample size required to determine whether a correlation coefficient differs from zero (α (two-tailed) =0.05; β=0.20; r=0.4) is 47. In accordance with the reviewer's remark, the above information has been supplemented as follows:
„This is a study that was conducted as part of obligatory practical classes in geriatric nursing with students in a strictly defined period of the academic year from March to June 2022. Out of all elderly patients staying at that time in three Departments of Internal Medicine of the Medical University of Lodz, Poland, 141 people (63 men and 78 women) elderly patients were selected who met the inclusion criteria of the study.”
Point 2: Section 2.2.1 - what is the rationale behind using SHARE-FI versus other frailty measures such as CFS, FRAIL or Fried's phenotype? Surprised that gait speed was not measured for these patients as it is important indicator of frailty?
Response 2: We are aware that there are several measures to assess the occurrence of frailty syndrome in older people. However, we chose the SHARE-FI calculator to assess the prevalence of the frailty syndrome because it is rated well in the available literature. Additionally, we decided to use Share-Fi because this measure has been used by us for many years. Our Department of Conservative Nursing at the Faculty of Health Sciences of the Medical University of Lodz participated in the process of validating this method, which is described in the following article:
Muszalik M, Borowiak E, Kotarba A, Puto G, Doroszkiewicz H, Kedziora-Kornatowska K. Adaptation and reliability testing of the SHARE-FI instrument for the assessment of risk of frailty syndrome among older Polish patients. Fam Med Prim Care Rev. (2018) 20:36–40. doi: 10.5114/fmpcr.2018.73702
This study is based on data collected as standard measurements during the patient's stay in the ward. Gait speed measurement is not a standard procedure in our hospital; therefore this variable was not taken into account. Our study is not a medical experiment; it is a typical observational study. Nevertheless, this is a valuable point and we will want to include this measurement in future research.
Point 3: Section 2.2.6 - I am surprised that ethics approval was not required for this study and yet the authors mentioned that patients signed informed consent form before procedures were performed. This brings me to mention lines 331-332. There's a typo for line 331. So shouldn't there be a statement for line 332 since informed consent was taken?
Response 3: Thank you for this comment. To clarify the above remark, we have added the following sentence:
„The proposal of this study did not require the approval of the Bioethics Committee, as it does not bear the hallmarks of a medical experiment.”
“Patients signed informed consent for all the diagnostic and therapeutic procedures during hospitalization.”
Point 4: Section 3 - I personally feel that there's not much added value to include table 1 as some of the comparisons made and results obtained are obvious especially height, weight and handgrip strength (eg. men would be taller, heavier and bigger handgrip strength than women). If table 1 is important, would suggest to remove height and weight and only include BMI. One surprising observation is that percentage of women with heart diseases are significantly higher than men. Is that a common trend in Poland or could it be due to sampling issue?
Response 4: Thank you very much for this comment. As suggested by the Reviewer, the suggested variables as well as some other less informative data were removed from the Table 1.
TABLE 1| General characteristic of the study population (n=141) according to sex.
Quantitative variables |
All N=141 |
Women N=78 |
Men N=63 |
Mann-Whitney U-test/Chi2 p-value |
|
|
Demography |
|
|
Age [years] |
73 (68-81) |
73 (67-81) |
73 (68-80) |
NS |
Education [n,%] Higher Secondary Vocational Primary |
21 (14.9) 61 (43.3) 31 (22) 28 (19.9) |
10 (12.8) 35 (44.9) 19 (24.4) 14 (18) |
11 (17.5) 26 (41.3) 12 (19.1) 14 (22.2) |
NS |
Place of living [n,%] Urban Rural |
118 (83.7) 23 (16.3) |
65 (83.3) 13 (16.7) |
53 (84.1) 10 (15.9) |
NS |
Living status [n,%] Alone With family |
65 (46.1) 76 (53.9) |
38 (48.7) 40 (51.3) |
27 (42.9) 36 (57.1) |
NS |
Marital status [n,%] married not married/single widowed |
62 (44) 14 (9.9) 65 (46.1) |
30 (38.5) 7 (9) 41 (52.6) |
32 (50.8) 7 (11.1) 24 (38.1) |
NS |
Economic status [n,%] very good good average bad |
9 (6.4) 56 (39.7) 75 (53.2) 1 (0.7) |
4 (5.1) 36 (46.2) 38 (48.7) 0 |
5 (7.9) 20 (31.8) 37 (58.7) 1 (1.6) |
NS |
Anthropometry and state of health |
||||
BMI [kg/m2] |
25.3 (22.6-29.7) |
25.3 (22.8-29.3) |
24.9 (21.8-32) |
NS |
IADL [number of points] |
24 (20-27) |
24 (20-26) |
24 (21-27) |
NS |
IADL Category [n,%] low medium high |
2 (1.4) 26 (18.4) 113 (80.1) |
1 (1.3) 14 (18) 63 (80.8) |
1 (1.6) 12 (19.1) 50 (79.4) |
NS |
Mobility [n,%] independent cart balcony crutches lying patient |
108 (76.6) 3 (2.1) 7 (5) 16 (11.3) 7 (5) |
59 (76) 2 (2.6) 5 (6.4) 8 (10.3) 4 (5.1) |
49 (77.8) 1 (1.6) 2 (3.2) 8 (12.7) 3 (4.8) |
NS |
HGS max. [kg] |
22 (18-30) |
20 (15-28) |
30 (20-35) |
0.0004 |
GDS [number of points] |
3 (1-7) |
3 (1-6) |
2 (1-7) |
NS |
GDS Category [n,%] normal mood risk of depressive symptoms severe depressive symptoms |
104 (73.8) 26 (18.4) 11 (7.8) |
59 (75.6) 14 (18) 5 (6.4) |
45 (71.4) 12 (19.1) 6 (9.5) |
NS |
Fraility score |
2.02 (1.02-3.1) |
2.22 (1.27-3.02) |
1.98 (0.5-3.2) |
NS |
Diseases |
||||
Number of diseases |
3 (2-5) |
3 (2-5) |
3 (2-5) |
NS |
Number of medications taken |
6 (3-8.5) |
6 (3-9) |
6 (3-8) |
NS |
Use of a hearing aid |
19 (13.5) |
7 (9) |
12 (19.1) |
NS |
Use of glasses |
112 (79.4) |
61 (78.2) |
51 (81) |
NS |
Heart diseases, n [%] |
42 (29.8) |
29 (38.2) |
13 (20.6) |
0.03 |
Hypertension, n [%] |
85 (60.3) |
52 (66.7) |
33 (52.4) |
NS |
High blood cholesterol, n [%] |
20 (14.2) |
8 (10.4) |
12 (19.1) |
NS |
Stroke, n [%] |
9 (6.4) |
6 (7.7) |
3 (4.8) |
NS |
Diabetes, n [%] |
42 (29.8) |
22 (28.2) |
20 (31.8) |
NS |
Respiratory system diseases, n [%] |
17 (12.1) |
4 (5.1) |
13 (20.6) |
0.005 |
Cancer, n (%) |
12 (8.5) |
5 (6.4) |
7 (11.1) |
NS |
Stomach or duodenal ulcer, n [%)] |
9 (6.4) |
6 (7.7) |
3 (4.8) |
NS |
Notes: Mann-Whitney U-test/Chi2/Fischer’s exact test were calculated.
Abbreviations: BMI, Body mass index;
IADL, instrumental activities of daily living;
HGS, Handgrip strength (kg);
GDS, geriatric depression scale;
NS, statistically non-significant difference.
As a matter of fact, the percentage of women with heart diseases was significantly higher than men due to the sampling issue. This was hospitalized group of patients and the prevalence of several diseases may be different as compared to random community-dwelling sample.
Point 5: Table 2 makes more sense for me to answer the research question. Once again, would suggest to remove height and weight.
Response 5: As suggested by the Reviewer, the suggested variables were removed from the Table 2.
TABLE 2 |Comparison of anthropometric variables, test results and prevalence of chronic diseases between Frail, Pre-frail and Control group
Quantitative variables |
Frail N=55 |
Pre-frail N=52 |
Control N=34 |
Kruskal-Wallis p-value |
Demography |
||||
Age [years] |
77(69-84) |
71.5 (68-78.5) |
69.5 (66-79) |
0.006b |
Men [n,%] |
19 (35) |
24 (46.2) |
20 (58.8) |
NS |
Education [n,%] Higher Secondary Vocational Primary |
11 (20) 17 (30.9) 12 (21.8) 15 (27.3) |
6 (53.8) 28 (3.9) 11 (21.2) 7 (13.5) |
4 (11.8) 16 (47.1) 8 (23.5) 6 (17.7) |
NS |
Place of living [n,%] Urban Rural |
44 (80) 11 (20) |
45 (86.6) 7 (13.5) |
29 (85.3) 5 (14.7) |
NS |
Living status [n,%] Alone With family |
31 (56.4) 24 (43.6) |
20 (38.5) 32 (61.5) |
14 (41.2) 20 (58.8) |
NS |
Marital status [n,%] married not married/single widowed |
14 (25.5) 7 (12.7) 34 (61.8) |
29 (55.8) 6 (11.5) 17 (32.7) |
19 (55.9) 1 (2.9) 14 (41.2) |
0.006 |
Economic status [n,%] very good good average bad |
5 (9.1) 26 (47.3) 24 (43.6) 0 |
4 (7.7) 21 (40.4) 26 (50) 1 (1.9) |
0 9 (26.5) 25 (73.5) 0 |
NS |
Anthropometry and state of health |
||||
BMI [kg/m2] |
25.3 (22.8-30.5) |
25.6 (21.9-29.3) |
24.9 (22.3-29.1) |
NS |
IADL [number of points] |
21 (17-25) |
24 (22.5-27) |
27 (25-27) |
0.00001 a.b.c |
IADL Category [n,%] low medium high |
2 (3.6) 17 (30.1) 36 (65.5) |
0 7 (13.5) 45 (86.5) |
0 2 (5.9) 32 (94.1) |
0.0075 |
Mobility [n,%] independent cart balcony crutches lying patient |
35 (63.6) 1 (1.8) 3 (5.5) 10 (18.2) 6 (10.9) |
40 (76.9) 1 (1.9) 4 (7.7) 6 (11.5) 1 (1.9) |
33 (97) 1(2.9) 0 0 0 |
0.019 |
HGS max. [kg] |
20 (13-25) |
25 (20-32.5) |
32 (24-42) |
0.00001 a.b.c |
GDS [number of points] |
5 (2-8) |
2.5 (1-6.5) |
2 (1-3) |
0.0015b |
GDS Category [n,%] normal mood risk of depressive symptoms severe depressive symptoms |
35 (63.6) 14 (25.5) 6 (10.9) |
40 (77) 10 (19.2) 2 (3.9) |
20 (85.3) 2 (5.9) 3 (8.8) |
NS |
Fraility score |
3.54 (2.69-4.15) |
1.68 (1.42-2.11) |
0.05 (-0.66-0.5) |
0.0001a.b.c |
Diseases |
||||
Number of diseases |
4 (3-6) |
3 (2-3.5) |
3 (2-5) |
0.0001b.c |
Number of medications taken |
8 (6-10) |
5 (2.5-7) |
5 (3-7) |
0.0003 b.c |
Use of a hearing aid |
9 (16.4) |
6 (11.5) |
4 (11.8) |
NS |
Use of glasses |
48 (87.3) |
39 (75) |
25 (73.5) |
NS |
Heart diseases, n [%] |
23 (41.8) |
13 (26) |
6 (17.7) |
0.039 |
Hypertension, n [%] |
34 (61.8) |
28 (53.9) |
23 (67.7) |
NS |
High blood cholesterol, n [%] |
11 (20) |
4 (7.84) |
5 (14.7) |
NS |
Stroke, n [%] |
7 (12.7) |
2 (3.85) |
0 |
NS |
Diabetes, n [%] |
19 (34.6) |
15 (28.9) |
8 (23.5) |
NS |
Respiratory system diseases, n [%] |
8 (14.5) |
4 (7.7) |
5 (14.7) |
NS |
Cancer, n [%] |
6 (10.9) |
5 (9.6) |
1 (2.9) |
NS |
Stomach or duodenal ulcer, n [%] |
5 (9.1) |
1 (1.9) |
3 (8.8) |
NS |
|
|
|
|
|
Notes: Kruskal-Wallis/Chi2/Fischer’s exact test were calculated.
a-Prefrail vs control
b-Frail vs control
c-Frail vs. Prefrail
Abbreviations: BMI, Body mass index;
IADL, instrumental activities of daily living;
HGS, Handgrip strength (kg);
GDS, geriatric depression scale;
NS, statistically non-significant differences.
Point 6: Table 3 - do this need to be included or can it be parked under Appendix?
Response 6: As for Table 3, we would like to present the correlations contained in it because they are statistically significant and very interesting. Table 3 is not large, therefore we would prefer it to remain in the main text. Furthermore, as on the Tables 1 and 2 the suggested variables were removed from the Table 3.
Point 7: Table 4 - would appreciate a conventional presentation of logistic regression analysis
Response 7: The presentation has been corrected. The logistic regression analysis in Table 4 is presented as follows:
TABLE 4| The odds ratio of FS occurrence in one-way analysis for selected variables
Quantitative variables |
Odds ratio |
95% confidence intervals |
p-value |
Women (reference group) |
1 |
|
|
Men |
0.50 |
0.25-1.01 |
NS |
Age [years] |
1.06 |
1.02-1.11 |
0.0036 |
Married (reference group) |
1 |
|
|
Not married/single |
3.43 |
1.03-11.44 |
0.0451 |
Widowed |
3.76 |
1.74-8.11 |
0.0007 |
Without heart diseases (reference group) |
1 |
|
|
Heart diseases |
2.46 |
1.17-5.16 |
0.0172 |
Number of diseases |
1.41 |
1.17-1.69 |
0.0003 |
BMI [kg/m2] |
1.04 |
0.98-1.11 |
NS |
IADL [number of points] |
0.82 |
0.75-0.90 |
0.0000 |
GDS [number of points] |
1.13 |
1.03-1.25 |
0.0096 |
HGS max. [kg] |
0.90 |
0.86-0.94 |
0.0000 |
Point 8: Figure 1 - the resolution of the plots are not great and the caption within each plot is a little confusing, can it be clearer?
Response 8: In accordance with the Reviewer's remark, the resolution and caption of the Figure 1 have been completed as follows:
FIGURE 1 | Box-plot graphs among Frail, Pre-frail and Control groups with the results of the Kruskal-Wallis test of age -A; Body Mass Index (BMI) - B; Instrumental Activities of Daily Living (IADL) - C; Geriatric Depression Scale (GDS) - D.
Point 9: Section 4 - would want to see the strength and limitations of the study and clinical implications of the findings from this study.
Response 9: The strength and limitations of the study and clinical implications of the findings have been added as follows:
„According to the obtained results, it seems that in hospitalized older adults the occurrence of FS depends on multiple factors and may be important for further prognosis. Especially important aspect of FS prevention seems caring for muscle strength. Previous and present data show that low HGS is a crucial element of FS and should be included as a standard screening test for older adults' care. Patients with a low score should be given special care and have a further diagnosis of the frailty syndrome. Undoubtedly, further studies are needed to investigate the potential possibilities of alleviating the symptoms of FS.
- Study limitations
This study has several limitations, including 1) the relatively small sample of participants, 2) inclusion of only the Caucasian race in the study, 3) no use of the gait speed method. It is worth emphasizing that the strength of this study is that it demonstrates a significant relationship between the FS and poorer HGS. It points to the need to introduce HGS measurement in older adults as a recommended component of the Comprehensive Geriatric Assessment in medical facilities. Bearing in mind these limitations, the present study requires caution in data interpretation and should be continued and confirmed in future studies in larger groups of seniors.”
Additionally, the linguistic correction has been implemented.
Detailed answers to all comments of the Reviewer were provided. The revised manuscript is attached. Adjustments made in the text have been highlighted. The text is prepared in Microsoft Word. Adjustments made in the text are visible through the use of the “Track changes” feature.
Yours sincerely,
Dr Izabela Kozicka
Medical University of Lodz
Department of Conservative Nursing Faculty of Health Sciences
Jaracza Street 63
90-251 Lodz
Poland

Reviewer 2 Report
I appreciate the opportunity to review the article on "Factors determining the occurrence of frailty syndrome in hospitalized older patients".
The topic is particularly interesting.
However, I would like to make a few suggestions:
Right in the abstract I suggest removing the word test group "The occurrence of FS in the tested group of patient", which is misleading.
Throughout the introduction it would be important to deepen the different dimensions of the syndrome. The references need to be updated.
In the methodology it mentions that no ethics committee was required. I can't understand why not?
The results are presented in a clear way.
In the discussion I would have liked to see a greater depth of the implications of these results, what implications they have. More than just describing the main results.
As well as, in the conclusion. what the authors intended to do with these results.
The references need to be updated, there are many publications on the topic that could improve the article.
Anyway I congratulate the authors on their choice of topic.
Author Response
Response to Reviewer’s 2 Comments
We would like to thank the Reviewer for the analysis of this manuscript and the valuable remarks. The manuscript has been corrected according to the Reviewer’s comments. Below we provide a summary of our responses to the main remarks:
Point 1: Right in the abstract I suggest removing the word test group "The occurrence of FS in the tested group of patient", which is misleading.
Response 1: Thank you very much for this comment. The change has been made in the text as follows:
„The occurrence of FS in the group of studied patients was related to: age (p<0.001), widowhood (p<0.001), comorbidities (p<0.001), heart diseases (p=0.04), more medications taken (p<0.001), lower FA (p<0.001), weaker HGS and depression (p<0.001).”
Point 2: Throughout the introduction it would be important to deepen the different dimensions of the syndrome. The references need to be updated.
Response 2: Thank you for this comment. The “Introduction” and “References” have been modified in this regard as follows:
“According to the definition proposed by L. P. Fried, it is a physiological syndrome characterized by a decrease in reserves and resistance to stressors.”
„In this classical definition, the FS includes parameters such as reduced muscle strength (handgrip strength less than 20% for gender norm and body mass index (BMI)), subjective fatigue, unintentional weight loss (at least 4.5 kg per year), slowed walking (less than 20% of the time-to-walk gender 4.572m norm) and low physical activity. Three of the above-mentioned criteria are sufficient for making a diagnosis. FS is usually preceded by prefrailty syndrome, which includes one or two symptoms, and identifies a group of elderly people with a significantly increased risk of developing FS.”
In accordance with the Reviewer's remark, the citation in the Introduction has been supplemented. The following citations have been added:
Fried, L.P.; Tangen, C.M.; Walston, J.; Newman, A.B.; Hirsch, C.; Gottdiener, J.; Seeman, T.; Tracy, R.; Kop, W.J.; Burke, G.; et al. Frailty in older adults: evidence for a phenotype. J Gerontol A Biol Sci Med Sci 2001, 56, M146-156, doi:10.1093/gerona/56.3.m146.
Donatelli, N.S.; Somes, J. What is Frailty? J Emerg Nurs 2017, 43, 272-274, doi:10.1016/j.jen.2017.03.003.
Clegg, A.; Young, J.; Iliffe, S.; Rikkert, M.O.; Rockwood, K. Frailty in elderly people. Lancet 2013, 381, 752-762, doi:10.1016/s0140-6736(12)62167-9.
Wang, M.C.; Li, T.C.; Li, C.I.; Liu, C.S.; Lin, W.Y.; Lin, C.H.; Yang, C.W.; Yang, S.Y.; Lin, C.C. Frailty, transition in frailty status and all-cause mortality in older adults of a Taichung community-based population. BMC Geriatr 2019, 19, 26, doi:10.1186/s12877-019-1039-9.
Point 3: In the methodology it mentions that no ethics committee was required. I can't understand why not?
Response 3: Thank you for this comment. This was explained as follows:
„The proposal of this study did not require the approval of the Bioethics Committee, as it does not bear the hallmarks of a medical experiment.”
“This study is based on data collected as standard measurements during the patient's stay in the ward.”
“Patients signed informed consent for all the diagnostic and therapeutic procedures during hospitalization.”
Point 4: In the discussion I would have liked to see a greater depth of the implications of these results, what implications they have. More than just describing the main results.
Response 4: Clinical implications have been added as follows:
„According to the obtained results, it seems that in hospitalized older adults the occurrence of FS depends on multiple factors and may be important for further prognosis. Especially important aspect of FS prevention seems caring for muscle strength. Previous and present data show that low HGS is a crucial element of FS and should be included as a standard screening test for older adults' care. Patients with a low score should be given special care and have a further diagnosis of the frailty syndrome. Undoubtedly, further studies are needed to investigate the potential possibilities of alleviating the symptoms of FS.
- Study limitations
This study has several limitations, including 1) the relatively small sample of participants, 2) inclusion of only the Caucasian race in the study, 3) no use of the gait speed method. It is worth emphasizing that the strength of this study is that it demonstrates a significant relationship between the FS and poorer HGS. It points to the need to introduce HGS measurement in older adults as a recommended component of the Comprehensive Geriatric Assessment in medical facilities. Bearing in mind these limitations, the present study requires caution in data interpretation and should be continued and confirmed in future studies in larger groups of seniors.”
Point 5: As well as, in the conclusion. what the authors intended to do with these results.
Response 5: Thank you for this comment. This information has been added:
“Previous and present data show that low HGS is a crucial element of FS and should be included as a standard screening test for older adults' care. Patients with a low score should be given special care and have a further diagnosis of the frailty syndrome. Undoubtedly, further studies are needed to investigate the potential possibilities of alleviating the symptoms of FS.”
Point 6: The references need to be updated, there are many publications on the topic that could improve the article.
Response 6: As mentioned in Point 2, the references have been updated. Thank you for this comment.
In accordance with the reviewer's remark, the citation in the Introduction has been supplemented. The following citations have been added:
Fried, L.P.; Tangen, C.M.; Walston, J.; Newman, A.B.; Hirsch, C.; Gottdiener, J.; Seeman, T.; Tracy, R.; Kop, W.J.; Burke, G.; et al. Frailty in older adults: evidence for a phenotype. J Gerontol A Biol Sci Med Sci 2001, 56, M146-156, doi:10.1093/gerona/56.3.m146.
Donatelli, N.S.; Somes, J. What is Frailty? J Emerg Nurs 2017, 43, 272-274, doi:10.1016/j.jen.2017.03.003.
Clegg, A.; Young, J.; Iliffe, S.; Rikkert, M.O.; Rockwood, K. Frailty in elderly people. Lancet 2013, 381, 752-762, doi:10.1016/s0140-6736(12)62167-9.
Wang, M.C.; Li, T.C.; Li, C.I.; Liu, C.S.; Lin, W.Y.; Lin, C.H.; Yang, C.W.; Yang, S.Y.; Lin, C.C. Frailty, transition in frailty status and all-cause mortality in older adults of a Taichung community-based population. BMC Geriatr 2019, 19, 26, doi:10.1186/s12877-019-1039-9.
Detailed answers to all comments of the Reviewer were provided. The revised manuscript is attached. Adjustments made in the text have been highlighted. The text is prepared in Microsoft Word. Adjustments made in the text are visible through the use of the “Track changes” feature.
Yours sincerely,
Dr Izabela Kozicka
Medical University of Lodz
Department of Conservative Nursing Faculty of Health Sciences
Jaracza Street 63
90-251 Lodz
Poland

Round 2
Reviewer 1 Report
Thanks for the responses to my questions.
Reviewer 2 Report
I thank the authors for their efforts in reviewing the revision suggestions